# Role of GALNT2 on Insulin Sensitivity, Lipid Metabolism and Fat Homeostasis

**DOI:** 10.3390/ijms23020929

**Published:** 2022-01-15

**Authors:** Alessandra Antonucci, Antonella Marucci, Vincenzo Trischitta, Rosa Di Paola

**Affiliations:** 1Research Unit of Diabetes and Endocrine Diseases, Fondazione IRCCS Casa Sollievo della Sofferenza, San Giovanni Rotondo (FG), 71013 Foggia, Italy; a.antonucci@operapadrepio.it (A.A.); a.marucci@operapadrepio.it (A.M.); 2Department of Experimental Medicine, Sapienza University, 00161 Rome, Italy

**Keywords:** ppGalNAc-T2, O-linked glycosylation, metabolic abnormalities, insulin resistance, atherogenic dyslipidemia, obesity, hyperglycemia

## Abstract

O-linked glycosylation, the greatest form of post-translational modifications, plays a key role in regulating the majority of physiological processes. It is, therefore, not surprising that abnormal O-linked glycosylation has been related to several human diseases. Recently, *GALNT2*, which encodes the GalNAc-transferase 2 involved in the first step of O-linked glycosylation, has attracted great attention as a possible player in many highly prevalent human metabolic diseases, including atherogenic dyslipidemia, type 2 diabetes and obesity, all clustered on the common ground of insulin resistance. Data available both in human and animal models point to GALNT2 as a molecule that shapes the risk of the aforementioned abnormalities affecting diverse protein functions, which eventually cause clinically distinct phenotypes (a typical example of pleiotropism). Pathways linking GALNT2 to dyslipidemia and insulin resistance have been partly identified, while those for type 2 diabetes and obesity are yet to be understood. Here, we will provide a brief overview on the present knowledge on *GALNT2* function and dysfunction and propose novel insights on the complex pathogenesis of the aforementioned metabolic diseases, which all impose a heavy burden for patients, their families and the entire society.

## 1. Introduction

O-linked glycosylation, the greatest diverse form of post-translational modifications, affects the folding, stability, processing and trafficking of most secreted proteins [1], thus playing a key role in regulating many physiological processes. It is, therefore, not surprising that abnormal O-glycosylation has been related to several human diseases [2]. *GALNT2* encodes the polypeptide N-acetyl-galactosaminyl-transferase 2 (ppGalNAc-T2, also referred to as GalNAc-T2) [3,4], which transfers N-acetyl galactosamine to the serine/threonine hydroxyl group of specific acceptor peptide motifs [5,6,7] in the first step of O-glycosylation. In the last few years, GALNT2 has attracted great attention as a possible player in many human metabolic abnormalities that all share the common ground of insulin resistance, including atherogenic dyslipidemia [8,9,10,11,12,13,14], type 2 diabetes [15,16] and obesity [17,18]. After very briefly introducing insulin resistance, its role in several highly prevalent human diseases and how abnormal insulin signaling may cause it, we will offer to the readers an overview on what is known about GALNT2 function and dysfunction, with the aim of providing novel insights on the complex pathogenesis of the aforementioned metabolic diseases.

## 2. Insulin Resistance and Metabolic Diseases

Insulin resistance is a complex and highly prevalent metabolic disorder, which is mainly due to the inability of multiple insulin target sites, including but not limited to skeletal muscle, liver, and adipose tissue, to respond to insulin. It is widely accepted that insulin resistance represents a common pathogenic ground for many additional disorders and highly prevalent diseases, comprising atherogenic dyslipidemia [19], type 2 diabetes, obesity and hypertension [19,20], all clustering in metabolic syndrome [21] and in the strictly related cardiovascular disease and kidney dysfunction [19,22,23,24,25], which are important and established causes of morbidity and mortality worldwide [26]. In all, insulin resistance is one of the main causes of suffering for many patients and their relatives while imposing a heavy burden on healthcare systems and the whole society. The heavy, vast, heterogeneous and deleterious impact of insulin resistance is not surprising considering the central role insulin is known to play in orchestrating several interwoven signals, constantly occurring in the body, which coordinate the ability of a variety of organs to sense and supplement the availability of energy molecules. Unraveling the several and complex molecular abnormalities underlying insulin resistance in various target tissues is an essential prerequisite for tackling the whole constellation comprising the aforementioned cardiometabolic abnormalities and is, therefore, timely needed.

Over the last few decades, several possible pathophysiological mechanisms leading to insulin-resistant individuals eventually developing the aforementioned metabolic and vascular disorders have been addressed. Just as an example, many studies have demonstrated that when pancreatic beta-cells fail to provide compensatory insulin secretion (chronic compensatory hyperinsulinemia response) [19,27], insulin resistance leads to a progressive loss of beta-cell mass and/or function and, thus, to a further impairment of insulin secretion. This altered interplay between insulin sensitivity and secretion, in turn, leads to and/or exacerbates lipid and carbohydrate substrate imbalance and is instrumental for the development not only of type 2 diabetes, but also atherogenic dyslipidemia (characterized by high triglycerides and low high-density lipoprotein (HDL)-cholesterol levels) and all the additional abnormalities of the constellation described above [19,27,28,29].

The intertwined risk factors shaping the risk of developing insulin resistance also draw a complex scenario that has yet to be fully clarified. In general, it is widely accepted that over both genetic predisposition and environmental factors, obesity and adipose tissue dysfunction are major movers of insulin resistance [30,31]. It is also well established that insulin resistance is the major, and probably the only, link between overweight/obesity and the numerous adverse clinical outcomes related to adiposity excess, including the aforementioned metabolic syndrome, cardiovascular disease, kidney dysfunction, fatty liver disease and also some cancers [32]. Keeping focused on adipose tissue homeostasis, which is one of the main objectives of this article, it is of note that adiposity excess has been largely demonstrated to eventually cause dysfunctional and insulin-resistant adipose tissue, which, in turn, as plasma lipid levels increase, leads to visceral and ectopic fat accumulation [33]. However, it is also true, and very interesting indeed, that adipose tissue dysfunctionalities have also been described in non-obese insulin-resistant individuals [34,35,36,37], thus leaving open the possibility that the relationships between insulin resistance and adipose tissue dysfunction are binary, with each of the two players representing the possible cause or possible effect of this association. Thus, understanding the mechanisms underlying adipose tissue dysfunction is also a prerequisite for tackling the whole constellation of abnormalities which recognize insulin resistance as a common pathogenic ground and is, therefore, an urgent need.

The first step of insulin action involves its binding to the insulin receptor at the plasma membrane, which consists of two extracellular α and two transmembrane β subunits [38,39,40]. Following the binding of insulin, the receptor’s β subunit becomes tyrosine phosphorylated and then activated [38,39,40]. This activation stimulates the receptor’s intrinsic tyrosine kinase activity, which catalyzes the phosphorylation of other proteins. These proteins include several insulin receptor substrates (IRS1, IRS2, IRS3 and IRS4) [38,39,40], which act as “docking molecules” for Src-Homology 2 domain-containing proteins, eventually enabling downstream signal transmission [38,39,40]. The effects of insulin on glucose metabolism are mediated by IRS1, which activates phosphatidyl inositol-3-kinase, which then produces phosphatidyl-inositol-3,4,5-triphosphate (PIP3) [38,39,40]. PIP3, in turn, activates the serine/threonine protein kinase beta Akt2, the central mediator of insulin’s effects on glucose and lipid metabolism, including glucose uptake and the synthesis of glycogen and fatty acids [38,39,40]. Several genes encode molecules that inhibit this pathway by various mechanisms, including a direct interaction with the insulin receptor (ectonucleotide pyrophosphatase/phosphodiesterase 1, *ENPP1*) [41,42,43,44,45,46] or Akt2 (tribbles pseudokinase 3, *TRIB3*) [47], tyrosine dephosphorylation of the insulin receptor (protein tyrosine phosphatase non-receptor type 1, *PTPN1*), serine phosphorylation of IRS1 (tumor necrosis factor, *TNF* and interleukin 6, *IL*6) and 5′ dephosphorylation of PIP3 (inositol polyphosphate phosphatase-like 1, *INPPL1*; also known as SH2-containing inositol phosphatase 2, *SHIP2*) [38,39,40,48,49]. In some individuals, insulin resistance and related traits are likely to be caused by abnormalities in this complex network that is modulated by both stimulatory and inhibitory molecules [38,39,48,49]. In addition, some studies point to insulin resistance at the level of the pancreatic β cells as an important determinant of impaired insulin secretion [50,51], the best predictor of future loss of glucose homeostasis [52,53]. Overall, abnormal insulin signaling, through a combined effect on both insulin resistance and deficient insulin secretion, becomes a key factor in the pathogenesis of type 2 diabetes, atherogenic dyslipidemia, metabolic syndrome and all related metabolic and cardiovascular abnormalities. Although our knowledge has improved over the last three decades [40], several gaps on the precise molecular events leading to the failure of insulin signaling still remain to be understood. Some novel cues linking insulin signaling, insulin resistance, atherogenic dyslipidemia and adipose tissue dysfunction will be provided in the following part of this article.

## 3. GALNT2 Gene, Protein Function and Targets

*GALNT2*, located on chromosome 1q42.13 [4], belongs to a large family of homologous genes coding in humans for 20 different glycosyltransferases all involved in O-glycosylation [54]. As indicated by intron/exon positioning analyses [54], *GALNT2* likely shares a common ancestor with the other *GALNT* genes [54] and exhibits more than 98% sequence similarity among species [3]. Of note, phylogenetic analyses suggest that the enzymatic function of ppGalNAc-T2 has also been highly conserved throughout evolution [55,56].

ppGalNAc-T2 participates in the initiation step of O-linked glycosylation [57,58] by covalently linking N-acetyl-D-galactosamine (GalNAc) to the hydroxyl group of serine or threonine on specific peptide acceptor motifs [5,6,7]. It is of note that in contrast to the other isoenzymes of the same family, which show distinct non-redundant substrate specificity [59,60,61], ppGalNAc-T2 serves a broad spectrum of target peptides [57], thus modifying a large number of substrates and contributing enormously to widen protein diversity [57]. In addition, *GALNT2* is expressed in nearly all cell types and tissues [62] and contributes to the function of several proteins in an organ-specific manner [9,54,63,64,65,66,67,68,69,70]. Based on these premises, it is not unexpected that *GALNT2*/ppGalNAc-T2 is considered by far the major contributor of the whole O-glycoproteome [57].

Only a few ppGalNAc-T2-targets have been identified so far. The most studied are angiopoietin-like protein 3 (ANGPTL3) [71,72,73], and apolipoprotein C3 (ApoC-III) [9,74], both impairing triglycerides (TG) clearance through inhibition of lipoprotein lipase (LPL) [75,76,77,78,79] and phospholipid transfer protein (PLTP) [9] that transfers phospholipids from TG-rich proteins to HDLs [80,81], which all point GALNT2 as a main actor on lipid metabolism. In addition, ppGalNAc-T2 glycosylates and affects turnover, steady-state levels and dimerization [64,65,66,67] of membrane receptors involved in several energy metabolic pathways and cell growth (i.e., β1-adrenergic [64], epidermal growth factor (EGF) [65,66], insulin-like growth factor 1 (IGF-1) receptors [67]), thus potentially affecting downstream events. Finally, ppGalNAc-T2 modulates the release of TNF-α [70], a central player of chronic inflammation, adipose cell metabolism and insulin resistance [82]. Based on these findings, it is conceivable that changes in *GALNT2*/ppGalNAc-T2 expression and/or function may contribute to shaping the risk of several metabolic abnormalities characterized by energy substrate imbalance, systemic low-grade inflammation and vasculature damages.

## 4. GALNT2 and Insulin Sensitivity

Evidence on the role of *GALNT2* as a mediator of insulin sensitivity comes from our studies on human liver cells (HepG2) as well as on mouse fibroblasts (3T3L1) during their differentiation into mature adipocytes [83,84]. The reduction in *GALNT2* expression in HepG2 cells impairs insulin signaling and action, as indicated by the reduced ability to stimulate insulin receptor, IRS-1 and protein kinase beta Akt2 phosphorylation [83] and to down-regulate the gluconeogenetic enzyme phosphoenolpyruvate carboxykinase (PEPCK) [83]. Coherently, data in pre-adipocytes show that *GALNT2* over-expression significantly improves insulin signaling and, in turn, stimulates adipocyte maturation and enlargement [84]. Unfortunately, the mechanism through which *GALNT2* affects insulin sensitivity is not yet known. In this regard, it is of note that in both HepG2 and 3T3L1 cells, the expression of *GALNT2* is inversely correlated with that of *ENPP1*, a well-established negative modulator of insulin signaling [41,42,43,44,45,46]. This correlation makes it possible to hypothesize that *ENPP1* down-regulation is likely to mediate, at least partly, the insulin-sensitizing effect of *GALNT2*. Whether ENPP1 is a primary ppGalNAc-T2 target or its modulation is secondary to upstream O-glycosylation events is not yet known. Taken as a whole, our data on two different insulin responsive cells do suggest that *GALNT2* is a positive modulator of insulin sensitivity.

A role of GALNT2 in modulating insulin sensitivity is also suggested by preliminary genetic studies indicating that *GALNT2* variability is associated with HOMA-insulin resistance index in women with polycystic ovary syndrome [85].

## 5. GALNT2 and Atherogenic Dyslipidemia

As said before, three main players in lipid metabolism have been validated as non-redundant substrates of ppGalNAc-T2. These are ANGPTL3 [71,72,73] and ApoC-III [9,74], which affect TG clearance, and PLTP [9] that acts on HDL-C clearance. Therefore, it is not surprising that studies on gene variability, gene expression changes and loss-of-function (LOF) mutations have pointed to GALNT2 as a shaper of both serum HDL-C and TG levels [8,9,10,11,12,13,14]. 

Extreme conditions of *GALNT2* loss of function (LOF), ranging from humans (homozygous individuals carrying the *GALNT2* p.Phe104Ser or p.Gln289 * mutations) to genetically modified nonhuman primates and mice, affect HDL-C consistently across species [9,86]. Notably, *GALNT2*-LOF also increased TG levels in monkeys and mice but not in humans [9], thus proposing that *GALNT2* modulates lipid metabolism in a species-specific manner.

Additionally, fine tuning of *GALNT2* affects lipid metabolism. In detail, data from genome-wide studies highlight that the common rs4846914 SNP at the *GALNT2* locus, which is associated with lower *GALNT2* expression in human liver [9], is also associated with increased TG and lower HDL-C levels [14], a combination known as atherogenic dyslipidemia, which is a marker of insulin resistance and an established risk factor for coronary artery disease (CAD) [81,87]. Along the same line, data in peripheral white blood cells (PWBC) from individuals with a wide range of metabolic clinical conditions show that *GALNT2* expression levels are correlated directly with HDL-C and inversely with TG serum levels [10]. Of note, *GALNT2* mRNA levels remain significantly associated with TG but no longer with HDL-C in conditional analyses including both lipid fractions, thus suggesting the association with HDL-C is mainly mediated by the very much expected strong and inverse correlation between HDL-C and TG levels [88]. Finally, the *GALNT2* promoter is hyper-methylated, a proxy of gene down-regulation, in patients with CAD and low HDL-C levels [19,89].

Overall, the available data depict a complex scenario compatible with the hypothesis that *GALNT2* exerts pleiotropic effects [88] on different serum lipid fractions by glycosylating alternative substrates, including ANGPTL3, ApoC-III and PLTP [88].

## 6. GALNT2, Type 2 Diabetes and Hyperglycemia

Studies in human PWBC show that *GALNT2* levels are reduced in obese patients with type 2 diabetes as compared to obese patients with no diabetes and even more to non-obese control individuals [15]. These human data resemble those reporting *GALNT2* down-regulation in liver of Goto-Kakizaki diabetic rats as compared to their normoglycemic counterparts [16]. The biology underlying these associations (e.g., is *GALNT2* down-regulation causing hyperglycemia or, vice versa, is hyperglycemia causing *GALNT2* down-regulation?) is clearly indicated by data in human cultured monocytes, where a high glucose concentration causes *GALNT2* down-regulation [15]. These data together with those on the direct relationship between *GALNT2* expression and insulin signaling [83,84] make it possible to hypothesize that hyperglycemia-induced *GALNT2* down-regulation is part of the mechanism underlying hyperglycemia-induced insulin resistance [90].

## 7. GALNT2 and Obesity

Given the well-established role of insulin sensitivity in promoting fat mass expansion, it is not surprising that GALNT2 over-expression in pre-adipocytes, by improving insulin signaling, stimulates adipocyte maturation [84] and leads to enlarged mature adipocytes characterized by small and clustered LD [91]. However, it cannot be excluded that GALNT2 also affects adipogenesis independently of insulin signaling, possibly mediating coordinated changes in key genes affecting adipocyte maturation (such as *Adig*, *Retn*, *Nr1h3*, *Ucp1*, *Slc2a4*, *Lipe*, *Fgf2* and *Tcs22d3*) [84]. Of note, *GALNT2* expression is increased in subcutaneous adipose tissue from obese individuals [17], thus making it possible to hypothesize a cause–effect relationship, with *GALNT2* expression changes contributing to the increase in adiposity. This hypothesis is further supported by models of *GALNT2*-LOF mutations in both cattle [18] and rodents [86] which, in addition to severe congenital abnormalities, show reduced body weight as compared to their wild-type counterparts [18,86]. 

## 8. Conclusions

Taken altogether, the data that we have reported here are quite consistent with the contribution of GALNT2 to several highly prevalent metabolic abnormalities sharing the common ground of insulin resistance, namely atherogenic dyslipidemia, type 2 diabetes and obesity. Unfortunately, they leave open the question on which mechanisms underlie some of the effects of GALNT2 on the aforementioned diseases. Pathways through which GALNT2 impairs lipid metabolism and insulin resistance have been partly identified, while those connecting GALNT2 to hyperglycemia/type 2 diabetes and obesity are only scarcely understood. With this caveat in mind, it is possible to envision a scenario in which GALNT2 operates through various mechanisms (Figure 1). The simplest, and probably too simple, picture is with GALNT2 changes inducing insulin resistance, which is pathogenic for virtually all the metabolic disorders mentioned above. Conversely, in a more complex and probably more realistic scenario, alternative pathogenic pathways, uniquely affecting different metabolic abnormalities, can be envisioned. This second hypothesis is facilitated by the knowledge that different degrees of *GALNT2* changes affect differently specific protein functions in an organ-specific manner, thus causing clinically distinct phenotypes (pleiotropism) [88]. A clear paradigm of this pleiotropic effect comes from data showing that while mild changes in ppGalNAc-T2 activity have been associated only with dyslipidemia [9,10,13,14], very rare and severe LOF homozygous mutations, other than heavily impairing lipid metabolism [9,86,92], cause the multisystem GALNT2-congenital disorder of glycosylation (CDG2T), which severely affects the central nervous system, muscle function, immunity control, several aspects of the endocrine system, and coagulation [18,86].

Overall, we must admit that, unfortunately, no single scenario is available within a solid framework for the many effects exerted by GALNT2 on the cluster of metabolic abnormalities, all sharing the common ground of insulin resistance. Several data indicate that they may actually represent different effects with different mechanisms (e.g., pleiotropism), although it is possible that, for at least some of them, a pathogenic link mediated by abnormal insulin signaling and insulin resistance does actually exist.

A further lack of knowledge concerns the possibility of modulating GALNT2 with the drugs currently in use for other purposes (i.e., drug repositioning). To the best of our knowledge, there are no in vivo or in vitro studies that have specifically investigated the effect of any drug on GALNT2 expression or ppGalNAc-T2 enzyme activity. Given the potential role of GALNT2 on the aforementioned severe and highly prevalent disorders, studies addressing the potential role of drugs already available today on GALNT2 would certainly be useful and are, therefore, timely.

In all, several studies both in humans and animal models point to GALNT2 as a contributing molecule to highly prevalent metabolic abnormalities, including atherogenic dyslipidemia, type 2 diabetes and obesity, which all impose a heavy burden for patients, their families and society as a whole. Conversely, our knowledge of the intimate mechanisms through which GALNT2 exerts these effects is still poor. A full understanding of GALNT2 function and dysfunction is, therefore, mandatory and will certainly improve our comprehension of the aforementioned metabolic diseases. Such knowledge gain is a prerequisite for discovering new pathogenic targets to be addressed with new therapies, so as to hopefully ameliorate the strategies currently available to treat patients with related insulin-resistance abnormalities.

## Figures and Tables

**Figure 1 ijms-23-00929-f001:**
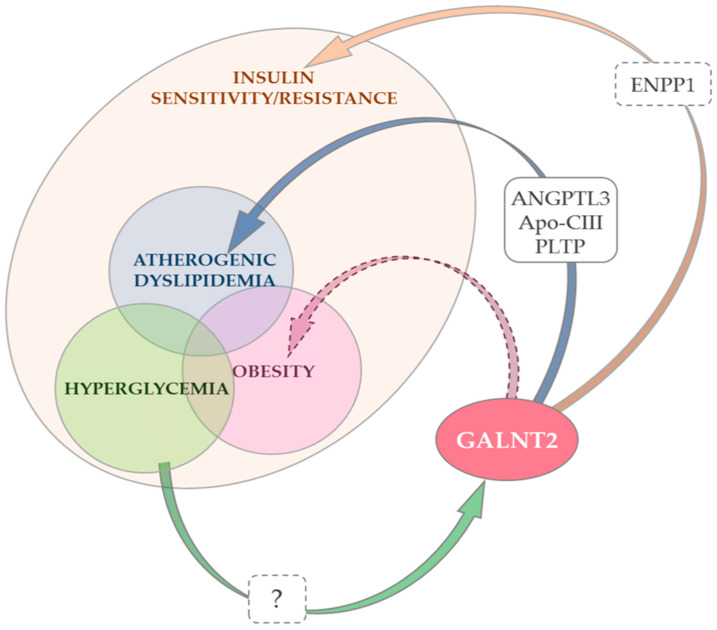
Changes in GALNT2 expression and/or function and insulin resistance, atherogenic dyslipidemia and obesity. Notes: GALNT2 promotes insulin action by down-regulating ectonucleotide pyrophosphatase/phosphodiesterase 1 (*ENPP1*) [83], a negative modulator of insulin signaling [41,42,43,44,45,46] (orange arrow). Whether ENPP1 is a primary ppGalNAc-T2 target or its down-regulation is secondary to upstream O-glycosylation events is unknown (dashed box). The role of GALNT2 down-regulation in atherogenic dyslipidemia (low HDL-Cholesterol and/or high triglycerides) is best understood. This effect is mediated by the O-glycosylation of three established targets of ppGalNAc-T2 [9] (the product of GALNT2) (blue arrow). Although no data are currently available, it cannot be excluded that some of the deleterious effects of GALNT2 down-regulation on dyslipidemia are mediated by the concurrent deleterious effect on insulin signaling which, in turn, causes insulin resistance and low HDL-cholesterol/high-triglyceride levels [19]. Experimental data suggest that the effect of GALNT2 on adipocyte maturation, adipogenesis and eventually obesity is at least partly due to its positive modulation of insulin signaling [84]. Whether GALNT2 also affects adipogenesis through other mechanisms, possibly mediating coordinated changes in key genes affecting adipocyte maturation [84], is an alternative possibility that deserves further studies to be addressed (purple dashed arrow). Finally, a high glucose concentration decreases *GALNT2* expression and this may play a role in hyperglycemia-induced insulin resistance (glucose toxicity) [15]. The mechanisms through which this down-regulation occurs are not known (green arrow, dashed box).

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
