# Peer review of "Role of GALNT2 on Insulin Sensitivity, Lipid Metabolism and Fat Homeostasis"

_ijms, 2022, doi:10.3390/ijms23020929_

Round 1
Reviewer 1 Report
The authors provide a global overview of the potential role of GALNT2 on insulin sensitivity, lipid metabolism, and fat homeostasis. The review is well written and provides an interesting insight while raising outstanding questions that remain to be addressed.
Minor comment:
Soil should be replaced with a more appropriate word.
Reviewer 2 Report
Comments and suggestions for authors:
In this review, the authors summarized and listed the role of GalNAc-transferase 2, their possible effect on insulin sensitivity, glucose and lipid metabolism and adipose tissue homeostasis. The individual parts of this review suggest possible impact of GALNT2 enzyme and of O-linked glycosylation process in the development of metabolic diseases, including atherogenic dyslipidemia, type 2 diabetes and obesity. The topic of the review is interesting, useful, clearly arranged with one figure and can have clinical implications. The whole review is well structured, although a bit difficult to read.
There are only couple of minor issues that have been mentioned in the „reviewer´s comments“.
Minor comments:
- First of all, with respect to clinical implication, is it anything known about the possibility of pharmacological affecting of GALNT2, e.g. by antidiabetic or antiobesitic drug? In my opinion, it would be appropriate and useful to add a short paragraph entitled „Implications and future perspectives“ to evaluate the opportunities and perspectives of pharmacological interventions.
- In line 99, please describe the expression „inositol3kinase“ as „inositol-3-kinase“.
- There are many abbreviations in whole review, therefore I encourage authors add the list of abbreviations, especially below the Figure 1. In addition, some abbreviations are not explain at all.
- In addition, the references are not consistent with the style of International Journal of Molecular Sciences. The authors should correct it according to authors quideline.
Author Response
In this review, the authors summarized and listed the role of GalNAc-transferase 2, their possible effect on insulin sensitivity, glucose and lipid metabolism and adipose tissue homeostasis. The individual parts of this review suggest possible impact of GALNT2 enzyme and of O-linked glycosylation process in the development of metabolic diseases, including atherogenic dyslipidemia, type 2 diabetes and obesity. The topic of the review is interesting, useful, clearly arranged with one figure and can have clinical implications. The whole review is well structured, although a bit difficult to read.
We thank the Reviewer for having found our work of interest as well as for providing us suggestions, which helped us improving the quality of the review.
There are only couple of minor issues that have been mentioned in the ‘reviewer´s comments’.
Minor comments:
Point 1: First of all, with respect to clinical implication, is it anything known about the possibility of pharmacological affecting of GALNT2, e.g. by antidiabetic or antiobesitic drug? In my opinion, it would be appropriate and useful to add a short paragraph entitled ‘Implications and future perspectives’ to evaluate the opportunities and perspectives of pharmacological interventions.
Response 1: We thank the Reviewer for having raised this important point. Unfortunately, there are no data currently available about the possibility modulating GALNT2 with anti-diabetes or anti-obesity drugs. However, we do agree it would be appropriate and useful to tackle this topic, which is now discussed on page 8, lines 1-7 of the revised version of our manuscript.
Point 2: In line 99, please describe the expression ‘inositol3kinase’ as ‘inositol-3-kinase’.
Response 2: The expression ‘inositol3kinase’ has been modified.
Point 3: There are many abbreviations in whole review, therefore I encourage authors add the list of abbreviations, especially below the Figure 1. In addition, some abbreviations are not explain at all.
Response 3: According to Reviewer's suggestion, a complete list of abbreviations has been provided on page 1.
Point 4: In addition, the references are not consistent with the style of International Journal of Molecular Sciences. The authors should correct it according to authors guideline.
Response 4: As kindly suggested by Reviewer, references have been corrected to be in accordance to the style of International Journal of Molecular Sciences.